# Decoding Electroencephalography Signal Response by Stacking Ensemble Learning and Adaptive Differential Evolution

**DOI:** 10.3390/s23167049

**Published:** 2023-08-09

**Authors:** Matheus Henrique Dal Molin Ribeiro, Ramon Gomes da Silva, José Henrique Kleinubing Larcher, Andre Mendes, Viviana Cocco Mariani, Leandro dos Santos Coelho

**Affiliations:** 1Industrial and Systems Engineering Graduate Program (PPGEPS), Pontifical Catholic University of Paraná (PUCPR), R. Imaculada Conceição 1155, Curitiba 80215-901, PR, Brazil; ramongs1406@gmail.com; 2Department of Mathematics, Federal University of Technology—Paraná (UTFPR), Via do Conhecimento, KM 01—Fraron, Pato Branco 85503-390, PR, Brazil; 3Mechanical Engineering Graduate Program (PPGEM), Pontifical Catholic University of Paraná (PUCPR), R. Imaculada Conceição 1155, Curitiba 80215-901, PR, Brazil; jose.kleinubing@pucpr.edu.br (J.H.K.L.); viviana.mariani@pucpr.br (V.C.M.); 4Department of Economics, Massachusetts Institute of Technology, 292 Main St, Cambridge, MA 02142, USA; andremendes1991@gmail.com; 5Department of Electrical Engineering, Federal University of Paraná (UFPR), R. Evaristo F. Ferreira da Costa 384, Curitiba 81530-000, PR, Brazil

**Keywords:** nonlinear system identification, machine learning, time-series modeling, differential evolution, electroencephalography signal response

## Abstract

Electroencephalography (EEG) is an exam widely adopted to monitor cerebral activities regarding external stimuli, and its signals compose a nonlinear dynamical system. There are many difficulties associated with EEG analysis. For example, noise can originate from different disorders, such as muscle or physiological activity. There are also artifacts that are related to undesirable signals during EEG recordings, and finally, nonlinearities can occur due to brain activity and its relationship with different brain regions. All these characteristics make data modeling a difficult task. Therefore, using a combined approach can be the best solution to obtain an efficient model for identifying neural data and developing reliable predictions. This paper proposes a new hybrid framework combining stacked generalization (STACK) ensemble learning and a differential-evolution-based algorithm called Adaptive Differential Evolution with an Optional External Archive (JADE) to perform nonlinear system identification. In the proposed framework, five base learners, namely, eXtreme Gradient Boosting, a Gaussian Process, Least Absolute Shrinkage and Selection Operator, a Multilayer Perceptron Neural Network, and Support Vector Regression with a radial basis function kernel, are trained. The predictions from all these base learners compose STACK’s layer-0 and are adopted as inputs of the Cubist model, whose hyperparameters were obtained by JADE. The model was evaluated for decoding the electroencephalography signal response to wrist joint perturbations. The variance accounted for (VAF), root-mean-squared error (RMSE), and Friedman statistical test were used to validate the performance of the proposed model and compare its results with other methods in the literature, including the base learners. The JADE-STACK model outperforms the other models in terms of accuracy, being able to explain around, as an average of all participants, 94.50% and 67.50% (standard deviations of 1.53 and 7.44, respectively) of the data variability for one step ahead and three steps ahead, which makes it a suitable approach to dealing with nonlinear system identification. Also, the improvement over state-of-the-art methods ranges from 0.6% to 161% and 43.34% for one step ahead and three steps ahead, respectively. Therefore, the developed model can be viewed as an alternative and additional approach to well-established techniques for nonlinear system identification once it can achieve satisfactory results regarding the data variability explanation.

## 1. Introduction

The monitoring of neural responses when a subject is exposed to a stimulus during medical procedures is commonly conducted through a non-invasive medical technique called electroencephalography (EEG). In this respect, the collected data compose a nonlinear system of observations, whose identification is challenging due to the high degree of nonlinearity. The most common methods for nonlinear system identification are the Nonlinear AutoRegressive Moving Average model with eXogenous inputs (NARMAX) [1,2], Volterra series [3], the Hammerstein model [4], the Wiener model [5], bilinear models [6], and state-space models [7]. Some aspects associated with these models, such as structure selection and determination, parameter estimation, and interpretability, can be characterized as challenges and disadvantages [8]. Alternatively, machine learning is a promising alternative [9].

Several contributions have been made concerning system identification for the human-brain-signal–movement interface with varying results. Vlaar et al. [10] verified that a linear model explains only about 10% of the EEG data variance in the same dataset as the one used in this study, leaving more than 80% as nonlinearities. A truncated Volterra series was used to model the data in this case. In the work of Tian et al. [11], a similar experiment was conducted on six individuals by recording the EEG response to wrist manipulation. A NARMAX model based on a hierarchical neural network was used, with a pre-processing step using independent component analysis. The proposed model achieved significantly better performance than a polynomial NARMAX model. Zhou et al. [4] studied the response of the human ankle, namely, its angle, to an electrical stimulus. The authors proposed a Hammerstein model based on artificial neural networks to predict the ankle angle. In this model, the parameters are estimated using a genetic algorithm. In Gu et al.’s study [12], a NARMAX model was used to create a dynamic model of cortical responses to mechanical wrist perturbations with one-step-ahead and three-step-ahead predictions.

Aljamaan [5] used a Wiener Box–Jenkins model in conjunction with a prediction error method for system identification in EEG signals as a response to wrist manipulation. The model performed one-step-ahead predictions. Van de Ruit [13] performed experiments to measure time-varying joint impedance on the wrist in response to perturbations in position. The authors used kernel-based regression and skirt-decomposition-based system identification. Westwick et al. [14] used data from the electromyogram (EMG) response to a rotational stimulus in the foot by evaluating the stretch reflex. The authors employed a separable least-squares optimization method to estimate the parameters of a Hammerstein cascade model, usually employed to model biological systems. Also, in relation to decoding motor tasks, Nicolas-Alonso et al. [15] used EEG data to classify the motor task that a test subject performs. The problem is divided into binary and multiclass classifications. Different pre-processing techniques, such as band-pass filtering, spatial filtering, and mutual-information-based best-individual feature selection, are used. The classification uses an ensemble of staked regularized linear discriminant analysis models. In the work of Dalhoumi and Montmain [16], a new method to calibrate models to classify users’ motor tasks is proposed, where multiple base learners are trained using data from previous subjects. The base learners are then weighted to form new predictions based on a few data points from the new user, reducing the calibration time.

Li et al. [17] used transfer learning to decode low-sampling-frequency surface EMG in a dataset of 53 different gestures of 10 subjects. Using a low sampling frequency allows for applying low-cost EMG sensors that can be used in robotic prosthesis control. In Lee et al.’s study [18], the weighted ensemble learning of convolutional neural networks (CNN) was used on a dataset consisting of nine types of arm movements from 12 subjects. The base CNNs were trained on data from different sessions but shared a joint weighted loss function in the training process. Therefore, it was possible to generalize the results, leveraging data from all sessions. Fagg et al. [19] suggested decoding arm movements in monkeys with electrode implants in the primary motor cortex area responsible for arm movement. The model consists of multiple signal-filtering methods and a linear filter decoder or a Wiener filter to predict the movement.

Considering the previous related works, there are limited studies regarding the use of artificial intelligence approaches to deal with nonlinear system identification. In this respect, there is an open question: what is the performance of ensemble learning methods compared with the current state of the art, such as NARMAX structures and other machine-learning models? Therefore, in this paper, we seek to provide answers to the previous question. An ensemble learning approach for system identification can be considered within the machine-learning field, such as the approach developed by Li et al. [20]. This methodology increases model accuracy by combining the predictions of several base learners through the weighted average rule in regression problems to solve the same problem [21]. The goal is for each model to learn some data patterns, and when their predictions are aggregated, an effective model is obtained [22,23,24]. A particular approach based on ensemble learning is stacked generalization (STACK). The STACK ensemble learning model executes the training process by layers and has produced accurate results, considerably lowering error values, in regression tasks in different fields of knowledge [25]. In this approach, base learners/models are trained in the first layer (layer-0, set of base learners), and their predictions are used as inputs in the subsequent layer [26]. Then, a meta-learner is trained to obtain the system’s final prediction in the next stage. This model will be able to deal with data nonlinearities since layer-0 models add different information from the predicted data for the meta-learner [27].

When aiming to increase the accuracy of some models, the hybridization of two or more methodologies is adequate, such as ensemble learning modeling and optimization (in general evolutionary algorithms or swarm intelligence approaches). Usually, the first is used for data modeling, and the second is used to select the ensemble’s hyperparameters. Considering the use of optimization methods in the metaheuristics field to tune hyperparameters, the main advantage is that an objective function will be optimized, allowing us to find suitable hyperparameters and obtain an adequate model, as can be observed in [28]. In this respect, differential evolution (DE) proposed by Storn and Price [29] has already been employed in several fields and achieved success in optimization tasks [30]. In this context, the Adaptive Differential Evolution with Optional External Archive [31,32] (JADE) algorithm should be highlighted. The optimization process becomes even more general when using an adaptive mechanism for the crossover and mutation operations, with no need to infer optimal values for those parameters, incorporating them into the optimization process. Once they are in the stacking ensemble learning model, the final performance depends on the meta-learner, and the suitable choice of hyperparameters can lead to better results, where optimization methods can be adopted.

Metaheuristics have been used with promising results for optimizing machine-learning model hyperparameters [33,34,35]. The gradient-free approach provided by JADE and other evolutionary algorithms makes them particularly well-suited for this type of task. Moreover, in most cases, these methods require defining initial settings to perform the optimization, interfering with the optimization process. To overcome that, we used JADE in this study because it is a self-adaptive version of DE that does not require defining parameters such as crossover and mutation probability. However, we also believe that other self-adaptive methods can be adopted for this purpose once our focus is not on the performance of JADE specifically but on how the accuracy of the stacking ensemble learning method can be improved when an optimizer method is used to tune the meta-learner hyperparameters to the system identification performed in this study. Additionally, evolving ensemble learning methods for free with evolutionary algorithms can be employed to obtain efficient predictive models [36].

Therefore, the objective of this paper lies in the proposition of a new hybrid framework by combining the STACK ensemble and JADE algorithm to realize the identification of a benchmark system related to cortical responses evoked by wrist joint manipulation [10,37] to make one-step-ahead predictions. The adopted benchmark has a single input and single output, and feature engineering was performed to improve the performance of the developed model. Since diversity plays a key role in ensemble modeling, a set of diversified models, varying in their structures, were used to compose STACK’s layer-0. The chosen models were eXtreme Gradient Boosting (XGBoost) [38] with a linear booster, a tree-based model, a Gaussian Process (GP) [39], a statistical model using a radial basis function kernel, Least Absolute Shrinkage and Selection Operator (LASSO) [40], a linear model with regularization, a Multilayer Perceptron (MLP) Neural Network [41] with one hidden layer, a neural network, and Support Vector Regression [42] with a radial basis function kernel (SVR), a model based on support vectors. All the models come from different families with distinctive characteristics, ensuring the much-needed diversity of the base learners.

In training, 6-fold cross-validation (CV) was used for base learners and the meta-learner. To define the layer-0 models’ hyperparameters, a grid search was adopted. The Cubist regression-rule-based model [43] was used as a meta-learner, and JADE was employed to obtain its hyperparameters.

This study mainly makes the following contributions: (i) A novel combination named JADE-STACK is proposed and applied to identify EEG signals; (ii) comparisons of the results obtained with the proposed approach and layer-0 models are conducted. At the same time, the performance of the proposed model is compared using variance accounted for (VAF) and root-mean-squared error (RMSE) with the results presented in the literature for the problem studied in one- and three-step-ahead EEG predictions; and (iii) a discussion about the feasibility of the proposed STACK for nonlinear system identification is presented.

The remainder of this paper is structured as follows. Section 2 introduces the benchmark adopted. Section 3 details the methods explored in this paper. Section 4 describes the proposed methodology. Section 5 presents the results and discussion. Finally, Section 6 concludes the paper and suggests future research.

## 2. Description of the Benchmark

The dataset used in this study is the first version of the benchmark proposed by [10], available at http://www.nonlinearbenchmark.org/#EEG (accessed on 29 July 2023), and refers to the response in the human cortex to robotic manipulation of the wrist joint. There were ten healthy volunteers (age range 22–25 years old, all right-handed). Participants were seated with their right forearm fixed to the arm support and their hand strapped to the handle of a robotic manipulator, as illustrated in Figure 1a, adapted from Vlaar et al. [10]. Perturbation was applied to the wrist by a robotic manipulator and is considered the input of the dataset.

The cortical activity was recorded using EEG. A headpiece with 126 electrodes arranged according to the 10–5 system [44] was used to measure the scalp potential. All signals were sampled at 2048 Hz with 136 channels. The signal acquisition process is further described in [37]. EEG data were pre-processed with a high-pass filter, independent component analysis was performed, and line noise at 50 Hz was removed in the frequency domain via the discrete Fourier transform and 100 Hz or higher frequencies. The signal was then re-sampled at 256 Hz (*N* = 256 samples). Further detailed pre-processing is found in [10]. For this system, the independent component of the EEG with the highest signal-to-noise ratio is used as the output at time *t* (y(t),t=0,…,1 s), and the handle angle when the wrist is stimulated is used as a system input at time *t* (u(t),t=0,…,1 s). Seven multisine perturbation signals are generated, and Figure 1b shows a one-second excerpt of one of the signals. Each participant was subjected to 49 different trials of 36 s each. Each trial utilized three different perturbations of the seven possible, cycling 3-by-3 throughout the 49 trials. Of the seven perturbation signals generated and applied for every participant, the first six were used as the training set, and one was used as a test or validation set. From every 36 s trial, 6 s was removed, as highlighted in Figure 1c, accounting for possible transient effects and leaving 30 s to be utilized.

## 3. Methods

This section presents the main aspects of the methods proposed in this paper. First, we present the STACK approach, followed by a description of the layers composing the final ensemble learning model.

### 3.1. Stacked Generalization

Ensemble learning, a methodology utilized in classification, regression, clustering, and nonlinear identification problems, aims to acquire a precise model [45]. This technique follows the divide-and-conquer principle by combining multiple base learners (weak learners) to construct an effective model, typically using the average rule for regression problems [22].

An alternative approach to ensemble learning is STACK-based ensemble learning, which enhances accuracy by integrating diverse base learners and utilizing multiple layers. Initially, base learners are trained in the first layer (layer-0), and their predictions are then utilized in subsequent layers. Subsequently, a meta-learner (layer-1) is trained using the predictions from the previous layer as inputs to generate final predictions. The effectiveness of this approach lies in its ability to capitalize on different models producing distinct predictions, from which the meta-learner learns.

The STACK ensemble learning method improves the forecasting output by reducing the variance in forecast errors or rectifying biases. Through this learning process, the results tend to converge toward an improved solution compared to using only the base learners [27]. However, one drawback of this approach is the challenge of selecting the appropriate number of base learners. Nevertheless, this issue can be addressed by employing readily available general-purpose machine-learning wrappers, such as caret for R [46], which facilitate the testing of numerous different base learners, as they offer a unified syntax.

Figure 2 illustrates the STACK ensemble learning methods for regression.

It is worth observing that STACK ensemble learning is an efficient tool to deal with different kinds of data features’ predictive contexts in some fields, such as financial [47], epidemiological [48], agribusiness [25], seismic [49], electrical power system [50], wind power [51], and photovoltaic solar energy [52] forecasting.

### 3.2. Models Used in STACK Methodology

This subsection introduces the base learners GP, SVR, LASSO, MLP, and XGBoost that compose STACK’s layer-0 and Cubist as the meta-learner.

A GP consists of a collection of random variables that follow a Gaussian distribution and are completely characterized by their mean and covariance (kernel) function [39]. The particular method in this paper adopts a GP with a radial basis function kernel. In this model, the sigma value serves as the sole hyperparameter.SVR involves identifying support vectors (points) near a hyperplane that maximizes the margin between two-point classes, which is determined by the difference between the target value and a threshold [42]. SVR incorporates a kernel, a function used to assess the similarity between two observations, enabling it to handle nonlinear problems. The radial basis function kernel is specifically employed in the model described in this paper. In this model, the sigma value serves as the sole hyperparameter.LASSO uses regularization terms to improve the model’s accuracy by sacrificing some bias to reduce the variance. Additionally, if there are some correlated predictors, LASSO selects the best set of them [40]. The hyperparameter of this model is the regularization value.The MLP is a feedforward neural network composed of one input layer (system’s input), one or more hidden layers, and one output layer (system’s outputs). In contrast to artificial neural networks without hidden layers, the MLP can solve nonlinearly separable problems [41]. In this paper, the MLP has a single hidden layer. The hidden layer activation function is the sigmoid function, while the output activation function is the identity. Using this configuration, the only hyperparameter for this model is the number of Hidden Units.XGBoost is based on a gradient boosting ensemble learning model and uses an additive learning strategy, which adds the best individual model to the current forecast model in the *i*-th prediction. A complexity term in the objective function is added to control the models’ complexity and helps smooth the final learned weights to prevent overfitting. Regularization parameters are used to avoid the estimation variance related to the base learner and to shrink them to zero [38]. The hyperparameters of this model are the number of boosting iterations, $L_{1}$ regularization, $L_{2}$ regularization, and the learning rate.Cubist is a rule-based model and operates using the regression tree principle [43]. Each regression tree and leaf is built with a rule associated with the information contained in each leaf. Given all rules, the final prediction is obtained as a rule’s linear combination. The concepts of committees (set of several Cubist results aggregated using the average) and the neighborhood are considered to build an accurate model. The hyperparameters of this model are the number of committees and neighbors.

A 6-fold cross-validation (CV) grid search was adopted to find base learners’ hyperparameters. This approach consists of evaluating all possible combinations of hyperparameters through an exhaustive search and determining the most suitable set of values. A grid search is a suitable approach for the stage in this paper since there are several base learners, and using optimization techniques would make the process computationally expensive. Once one meta-learner exists, the JADE approach finds the Cubist hyperparameters.

### 3.3. Adaptive Differential Evolution with Optional External Archive

Different DE-based approaches have shown themselves to be capable of achieving good results in various optimization tasks, as presented in [53], incorporating a differential-evolution-based search into the Downhill Simplex method. The JADE algorithm improves DE optimization, which implements a new mutation strategy named “DE/current-to-pbest/1” with the optional external archive and updates control parameters in an adaptive manner [31,32].

According to [31], the step-by-step process employed in the JADE approach is described as follows:Initialization: The first population is randomly initialized according to
(1)$$x_{ij}^{0}=x_{j_{UB}}+rand_{j}\left(x_{j_{UB}}-x_{j_{LB}}\right),$$
where *i* = 1,…, NP, j=1,…,d,
LB and UB are the lower and upper boundaries of the search space for the parameter *j*, $rand_{j}$ denotes a uniform distribution within [0,1], x denotes a candidate solution (or agent) in the population, and NP is the size of the population. After that, the algorithm enters a loop of evolutionary operations: mutation, crossover, and selection.Mutation: Considering that the mutation strategies of classical DE are fast, but the convergence can be less reliable, the JADE approach adopts the mutation vector as follows. With P as the set of current solutions and A as the poor solutions archived, the mutation vector $v_{i,g}$ can be written as
(2)$$v_{i,g}=x_{i,g}+F_{i}\times \left(x_{best,g}^{p}-x_{i,g}\right)+F_{i}\times \left(x_{r1,g}-{\tilde{x}}_{r2,g}\right),$$
where $x_{i,g},x_{r,1,g}$, and $x_{best,g}^{p}$ are selected from P, and ${\tilde{x}}_{r2,g}$ is randomly selected from P∪A. Moreover, the mutation factor $F_{i}$ associated with $x_{i}$ is restarted adaptively in each generation (iteration). In this context, the generation of the $F_{i}$ parameter can be described as
(3)$$F_{i}=randc_{i}\left(\mu_{F},0.1\right),$$
where $randc_{i}\left(\mu_{F},0.1\right)$ refers to a Cauchy Distribution with a location parameter of $\mu_{F}$ and a scale parameter of 0.1. However, if the value of $F_{i}$ is less than or equal to 1, the distribution is truncated to 1. Conversely, the distribution is regenerated if $F_{i}$ is greater than or equal to 0. The set $S_{F}$ represents all the successful mutation factors in generation *g*. The initial value of the location parameter is set to 0.5. At the end of each generation, this parameter is updated as
(4)$$\mu_{F}=\left(1-c\right)\times \mu_{F}+c\times mean_{L}\left(S_{F}\right),$$
where $mean_{L}$ is the Lehmer mean, and *c* is a constant value between 0 and 1. Second, according to [54], the purpose of this archive is to counteract the greedy nature of the current best solution within the mutation phase, preventing premature convergence.Crossover: In the crossover step, a binomial operation forms the trial/offspring vector or individual *i* in generation *g* and dimension *d*, $u_{i,g}=\left\{u_{1,i,g},u_{2,i,g},\ldots ,u_{d,i,g}\right\}$, where
(5)$$u_{j,i,g}=\left\{\begin{array}{cc} v_{j,i,g} & \mathrm{if}\;{\mathrm{rand}}_{j}\left(0,1\right)\leq CR_{i}\;\;\mathrm{or}\;j=j_{\mathrm{rand}}\\ x_{j,i,G} & \mathrm{otherwise}, \end{array}\right.$$
where $rand_{j}\left(0,1\right)$ is a uniform number chosen randomly between these boundaries, and $j_{rand}$ is an integer randomly chosen from 1 to *d* and is newly generated for each *i*. In this respect, $CR_{i}$ is the crossover probability of each individual $x_{i}$ and is independently generated according to
(6)$$CR_{i}=randn_{i}\left(\mu_{CR},0.1\right),$$
where randn represents a normal distribution with a mean of $\mu_{CR}$ and a standard deviation of 0.1. The values generated from this distribution are then truncated to the range of [0, 1]. To adaptively update the crossover probability, the set $S_{CR}$ is introduced, which consists of successful crossover probabilities $CR_{i}$ in generation *g*. In the first generation, the mean is initialized to 0.5. Then, in each subsequent generation, the mean is calculated using
(7)$$\mu_{CR}=\left(1-c\right)\times \mu_{CR}+c\times mean_{A}\left(S_{CR}\right),$$
where $mean_{A}$ is the usual arithmetic mean.Selection: Finally, it is necessary to find the new population by evaluating the elements ($u_{i,g}$) with the cost function. In this respect, if the value of the evaluated cost function for the offspring is better than the value for the parent ($x_{i,g}$), the new element ($x_{i,g+1}$) will be the mutated vector. Otherwise, it will be equal to the early parent.

Figure 3 presents the JADE algorithm’s workflow.

## 4. JADE-STACK Algorithmic Description

The main steps adopted for the identification are described as follows:1.Initially, six perturbation signals of the seven available for each participant are used as the training set. Every participant’s seventh and last signals are used as a test set. This split is applied based on the initial evaluation of the dataset reported in [10,11] so it will be possible to compare the results. Moreover, considering the first six perturbation signals, a six-fold CV is used to train and validate the proposed model. Consequently, according to the proposal by [10], system identification is made for each participant individually one step ahead and three steps ahead, as suggested in [11]. Some features are obtained based on lagged input in one instant [u(t−1)]. The mean, standard deviation, and skewness in a window of three observations are defined. The difference between the lagged inputs in one and two instants is obtained. Finally, ${\left[u\left(t-1\right)\right]}^{3}$ and the logarithm of u(t−1) are also calculated to be used as input features.2.In sequence, the models described in Section 3.2 are trained according to the structure
(8)$$\begin{array}{cc} y(t)= & f\{y\left(t-1\right),\ldots ,y\left(t-n_{y}\right),\\ \; & u\left(t-1\right),\ldots ,u\left(t-n_{u}\right),u_{F}\left(t-1\right)\}+\epsilon \left(t\right), \end{array}$$
where *f* is a function used to map data, y(t) and u(t) are the output and input of the system at time *t* (t=1,…,256), $n_{y}=5$ is the maximum lag for outputs, $n_{u}=3$ is the maximum lag for inputs, $u_{F}\left(t-1\right)$, (F=1,…,6) are the features based on u(t−1) described in step 1, and ϵ(t) is the residual. A six-fold CV is adopted in this process, and the set of hyperparameters is defined for each model using a grid search. The obtained predictions compose STACK’s layer-0.3.In the meta-learner training, the JADE approach is used to find the Cubist hyperparameters based on the maximization of variance accounted for (*VAF*):
(9)$$VAF=100\times \left\{1-\frac{var[y\left(t\right)-\hat{y}\left(t\right)]}{var[y(t)]}\right\},$$
where var is the data variance, $\hat{y}\left(t\right)$ is the predicted output, and y(t) is the observed output. The Cubist hyperparameters are integer variables named the number of committees and neighbors. During the optimization process, real values are rounded to the next integer value. Indeed, the search space is defined between 0 and 60 for committees and 0 and 20 for neighbors. Additionally, 10 iterations (generations) and a population equal to 10 are used in JADE optimization. Once the hyperparameters are obtained, the meta-learner is trained again using a six-fold CV, and final predictions are obtained.Moreover, the *RMSE* is also computed as described in (Equation 10):
(10)$$RMSE=\sqrt{\frac{1}{n}\sum_{i=1}^{n}{\left(y_{i}-{\hat{y}}_{i}\right)}^{2}},$$
where $y_{i}$ and ${\hat{y}}_{i}$ are the *i*-th observed and forecast values, respectively, and *n* is the size of the time series.4.Finally, the results obtained in steps 2 and 3 are utilized to assess the statistical differences between the compared models using the variance-accounted-for (*VAF*) criterion. The Friedman test can be employed [55,56] to determine whether there is a significant difference among more than two models. This test helps determine whether a set of *k* models (greater than two) exhibit statistically significant differences in their results. The test statistic, denoted as FD, is calculated as follows:
(11)$$FD=\frac{12n}{k(k+1)}\left[\sum_{j}R_{j}^{2}-\frac{k{\left(k+1\right)}^{2}}{4}\right]∼\chi_{k-1}^{2},$$Here, FD follows a chi-squared distribution with k−1 degrees of freedom, based on *n* observations and *k* groups. The null hypothesis assumes that no difference exists between the results of the different groups.Subsequently, if the null hypothesis is rejected, a post hoc test is necessary to identify which groups have distinct results. In this case, the Nemenyi multiple-comparison test can be employed. This test determines the critical difference (CD), defined as
(12)$$CD=\frac{q_{\infty ,k,\alpha }}{\sqrt{2}}\sqrt{\frac{k(k+1)}{6}},$$
where CD represents the critical difference required to determine whether there is a significant difference in the measures of the groups. $q_{\infty ,k,\alpha }$ corresponds to the studentized range statistic, *k* represents the number of algorithms, α signifies the significance level, and *n* denotes the number of samples. If the absolute difference in rank sums $|R_{i}-R_{j}|$ is greater than CD, it indicates a significant difference between the results of algorithms *i* and *j* [57]. Thus, the Friedman and the Nemenyi multiple-comparison tests are applied to compare the errors obtained from all proposed algorithms.

Figure 4 presents the framework of the proposed optimized STACK. The results presented in Section 5 were generated using the caret package [46] of R software version 4.3.1 [58]. Table 1 shows the hyperparameters selected for each adopted model. Additionally, when the DE method was used to tune the meta-learner hyperparameters, the population size, crossover ratio, and mutation factor were equal to 100, 0.5, and 0.2, respectively. For the genetic algorithm (GA) method, the population size, mutation probability, and crossover probability were equal to 100, 0.3, and 0.8, respectively. In the same vein, for Particle Swarm Optimization (PSO), the swarm size, particle velocity, cognitive coefficient, social coefficient, and inertia weight were defined as equal to 100, 2, 2, 2, and 0.729, respectively.

Furthermore, Table 2 presents the calculation times of the training process of the proposed approach and the base models. It is important to note that although the average processing time of the JADE-STACK model appears to be longer, it is not designed for real-time learning. This means the model requires approximately 20 min to train, but once trained, it can be used for as long as it maintains accuracy. Additionally, the prediction step for the target dataset is almost instantaneous. Therefore, the training process should not pose any issues or limitations for its implementation in a production system.

## 5. Results and Discussion

Table 3 and Table 4 present the VAF and RMSE for ten participants in the forecasting horizon of one and three steps ahead, respectively, as proposed in [11]. Additionally, the average and standard deviation (Std) of the proposed model are compared with the results from the literature for the same dataset.

For the VAF comparison in one-step-ahead predictions, the optimized STACK improves the VAF by 1.51%, 8.48%, 2.85%, 5.50%, and 7.22% compared to LASSO, GP, MLP, SVR, and XGBoost, respectively. For the VAF in terms of three-step-ahead predictions, the proposed model can improve the explained variability by 1.22%, 13.66%, 11.64%, 9.43%, and 15.04% compared to LASSO, GP, MLP, SVR, and XGBoost, respectively. When we are dealing with the RMSE criterion, similar results are achieved. These results indicated that the proposed approach could learn the data behavior better than the other models. Additionally, for short-term forecasting, it can be stated that the proposed model can outperform the other compared methods regarding accuracy once it better explains the data variance.

As pointed out by [12], multi-step-ahead predictions are still challenging, especially for cortical activity. The recursive strategy used to perform multi-step-ahead predictions can be well suited. However, its use implies carrying the errors from the previous steps into the following predictions, making every step more susceptible to errors. This explains why the three-step-ahead predictions had a smaller VAF and a higher RMSE and shows how these predictions can be challenging for EEG signals.

The proposed framework achieves better results regarding accuracy than those obtained by [10,11] in the one-step-ahead prediction when the NARMAX-Hierarchical Neural Network (NARMAX-HNN) and Volterra series were used. JADE-STACK was able to identify the data with greater accuracy and fewer parameters. In [11], NARMAX-HNN has 19 parameters, the Volterra series has 46 parameters, while the model proposed in this paper has 11 hyperparameters for one step ahead and three steps ahead. Furthermore, considering the NARMAX-Polynomial (NARMAX-P), the proposed JADE-STACK is competitive and achieves better accuracy regarding VAF since a similar proportion of variability is explained with fewer parameters than the 24 needed for NARMAX-P. Also, the improvement over state-of-the-art methods ranges from 0.6% to 161% and 43.34% for one step ahead and three steps ahead, respectively. The improvement in VAF achieved by the proposed JADE-STACK is 2.4%, 0.6%, 120.6%, and 161% compared with the VAF of NARMAX-HNN, NARMAX-P, Volterra_1_, and Volterra_2_ for one step ahead. Alongside that, for three steps ahead, the improvement in VAF obtained with JADE-STACK compared with NARMAX-P is 43.34%. However, the VAF of NARMAX-HNN is 2.67% better than that of the proposed method. In terms of three steps ahead, considering the VAF criterion, the proposed model can achieve better results than NARMAX-P and is similar to NARMAX-HNN. Compared with these approaches, the proposed model seems more suitable for identifying a system composed of cortical activities since only 11 parameters are used (9 for base learners and 2 for the meta-learner). An important challenge with these data is the nonlinearities therein. As stated by [10], a simple linear model could only explain about 10% of the data variance, leaving the remaining data as nonlinearities. The machine-learning models, combined with the chosen features, show values of VAF that are significantly higher than 10%, as shown in Table 3 and Table 4.

When the proposed model results are compared with the DE-STACK model, it is possible to observe that using the JADE optimizer leads to better results by finding suitable hyperparameters for the meta-learner. Indeed, the improvement achieved by the proposed model is 1.70% and 2.86% in terms of one- and three-step-ahead predictions, respectively. Similar results are obtained by comparing JADE-STACK with GA-STACK and PSO-STACK. These results can be attributed to the fact that the JADE approach tunes its hyperparameters during the evolutionary process. In contrast, the DE, GA, and PSO parameters (crossover and mutation probability) are tuned by trial and error.

Figure 5, Figure 6, Figure 7, Figure 8, Figure 9, Figure 10, Figure 11, Figure 12, Figure 13 and Figure 14 show the comparison between observed and predicted values for all ten participants, as well as the residuals of predictions (difference between the observed and predicted values), in the test set (trial seven), according to the proposed JADE-STACK model in the task of predicting values one step ahead and three steps ahead. textcolorredThe horizons were based on the benchmark problem, so it would be possible to compare the results with the models proposed previously using the benchmark detailed in this paper. As pointed out by Tian et al. [11], dynamic brain activity is on the order of milliseconds, and the three steps ahead represent 12 ms ahead for EEG oscillations. Therefore, it is suitable for testing the proposed and compared frameworks. Moreover, further studies can be considered to introduce the concept of free-run for EEG prediction, as pointed out by Ayala and Coelho [59]. While the proposed model does not offer significant improvements for three-step-ahead predictions compared to the existing literature, the results are at least similar in terms of accuracy and consistency. However, most machine-learning problems do not have a clear all-purpose model to solve them. As stated by the No Free Lunch theorem [60,61], the engineer has the task of testing, identifying, and selecting the best tool to solve the problem at hand. Therefore, we can consider the evaluation of the proposed model valid. Indeed, the presented results are related to the performance of JADE-STACK for trial seven after performing a six-fold CV for each participant over the first six trials. In most cases, the proposed framework could capture the data variability and learn the data behavior by showing a similar pattern between observed and predicted values. In the Kolmogorov–Smirnov test (*K* = 0.0322, *p*-value = 0.7558), the residuals are white noise.

According to the Friedman test, there is a significant difference, in a statistical sense, between the VAF for all participants between the proposed and compared models for predicting one step ($\chi_{10}^{2}=93.29$, *p*-value = $1.22\times 10^{-15}$) and three steps ahead ($\chi_{8}^{2}=42.98$, *p*-value = $8.83\times 10^{-7}$). Figure 15 and Figure 16 depict the CD plot based on the Nemenyi test. When a line does not join two algorithms, there is a difference between their evaluated measures. When comparing the proposed model with others in the literature, it achieves the best results for one-step-ahead predictions and is the second-best for three-step-ahead predictions. Even though the errors of JADE-STACK and some models are statistically the same, the results obtained using the two models are different. Minor differences exist and should be considered [62].

## 6. Conclusions and Future Research

This paper proposes a novel combination of machine-learning and non-parametric regression models to build a STACK model to identify a system related to cortical responses extracted through EEG. JADE optimization was employed to tune the Cubist hyperparameters by maximizing the VAF. The proposed ensemble uses LASSO, GP, MLP, SVR, and XGBoost as STACK’s base learners and Cubist as the meta-learner. It also compared the performance of the proposed model with the base-learner performance and results reported in the literature from [10,11] for the analyzed benchmark in the context of one-step-ahead and three-step-ahead predictions of EEG signals.

The proposed optimized STACK outperforms the base learners and most of the reported methods from the literature proposed by [10,11], as evaluated VAF. These results show that hybridizing the STACK ensemble methodology with metaheuristics for system identification is a suitable tool since the obtained model can adequately capture data variability. We showed that our proposed STACK model is appropriate for this task. By using predictions of layer-0 models, the meta-learner learns the data behavior, accommodates data nonlinearities, and can obtain predictions similar to observed values. Also, it is crucial to notice that even single models, such as LASSO, had good performance considering VAF when compared with other models in the literature, which shows that even simpler machine-learning regression techniques can be appropriate approaches for system identification.

Despite the proposed framework achieving satisfactory results in terms of VAF, some limitations of the developed model can be identified. First, when we deal with the stacking ensemble approach, the selection of base and meta-models is challenging since there are no guidelines regarding the number of base models to be used and how many layers should be applied. Also, whether there is a better choice for a meta-model remains an open question. Third, feature selection is an essential step of system identification. However, this was not the focus of this study, despite being an essential step of data modeling.

The following future research is intended: (i) evaluate the proposed model using a free-run simulation [59]; (ii) adopt multi-objective optimization for hyperparameter tuning and feature selection using principal component analysis and independent component analysis; (iii) compare the proposed machine-learning model with variants of the high-order autoregressive models; (iv) to evaluate the use of L-SHADE [63] (Success-History-based Adaptive DE algorithm with Linear Population Size Reduction) and EPSDE [64] (DE Algorithm with Ensemble of Parameters and Mutation and Crossover Strategies), as well as swarm-based methods, such as the chaotic particle swarm optimizer in [65], the particle swarm used in [66] with a quasi-Newton local search, owl optimization [67], falcon optimization [68], the Jaya optimization algorithm [69], and cheetah optimization [70]. These variations have shown promising results in different areas and could also be used for system identification and hyperparameter tuning. Indeed, this proposed future research will help develop a more complete and effective model for the adopted dataset. Also, the intention of employing these suggestions is to develop an enhanced ensemble learning model to be applied in system identification.

Future research will approach high-order autoregressive (AR) models in comparison with machine-learning models for different step-ahead prediction horizons.

## Figures and Tables

**Figure 1 sensors-23-07049-f001:**
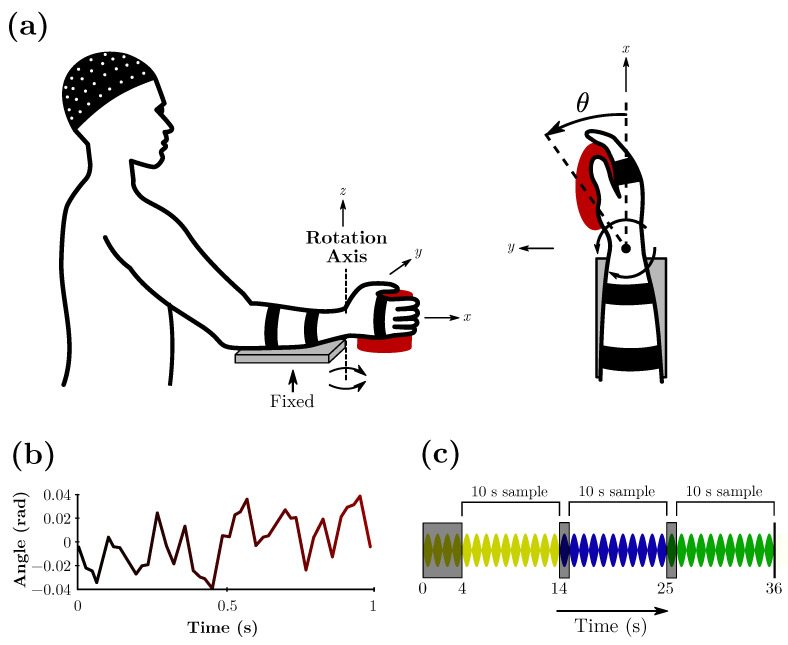
(**a**) Experimental setup for a specific participant. (**b**) Example of one second of the perturbation signal. (**c**) Representation of perturbations in a 36 s trial. The dark parts are removed to account for transient effects. Each color represents a different perturbation signal, and each lobe represents one second of the experiment.

**Figure 2 sensors-23-07049-f002:**
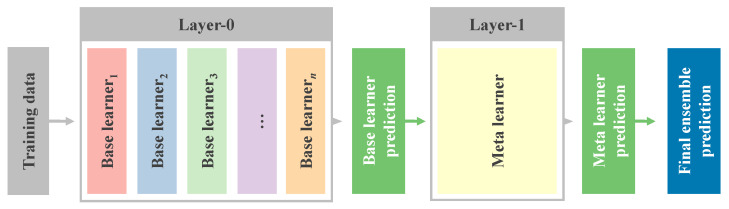
STACK ensemble learning workflow.

**Figure 3 sensors-23-07049-f003:**
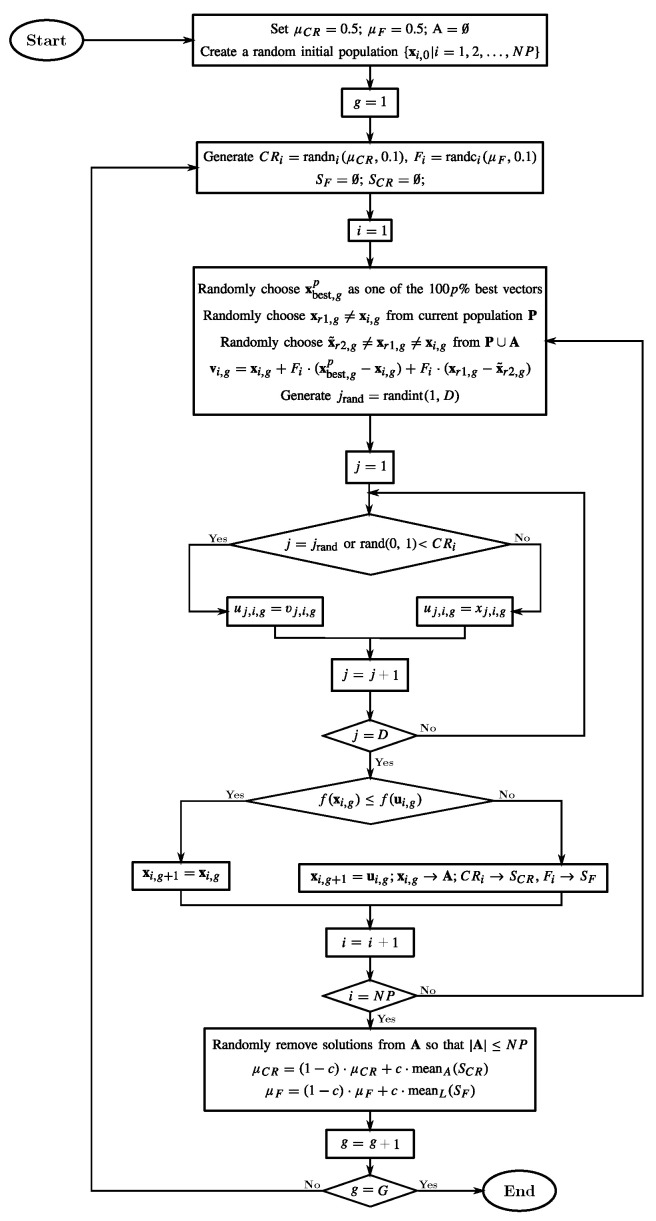
Flowchart of the JADE algorithm.

**Figure 4 sensors-23-07049-f004:**
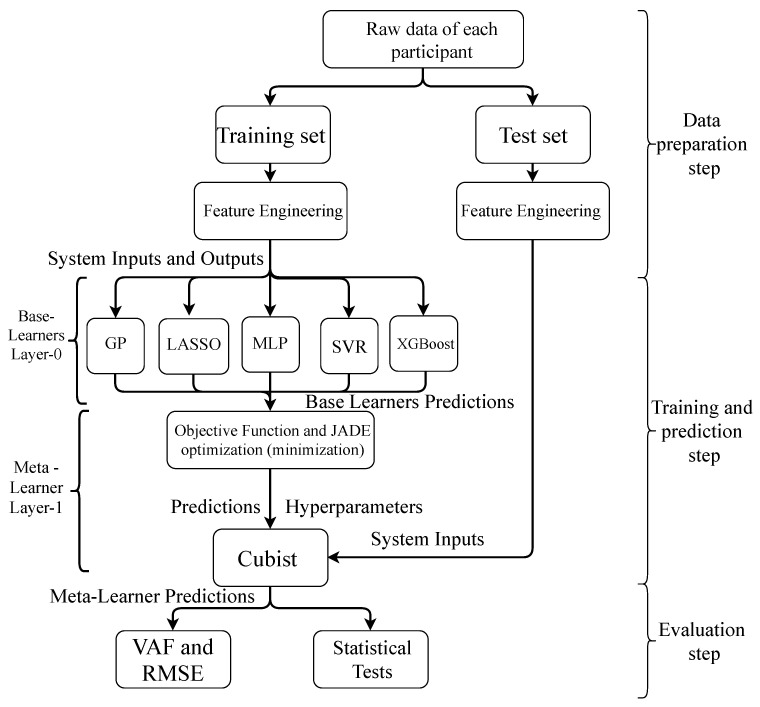
Framework of the JADE-STACK model.

**Figure 5 sensors-23-07049-f005:**
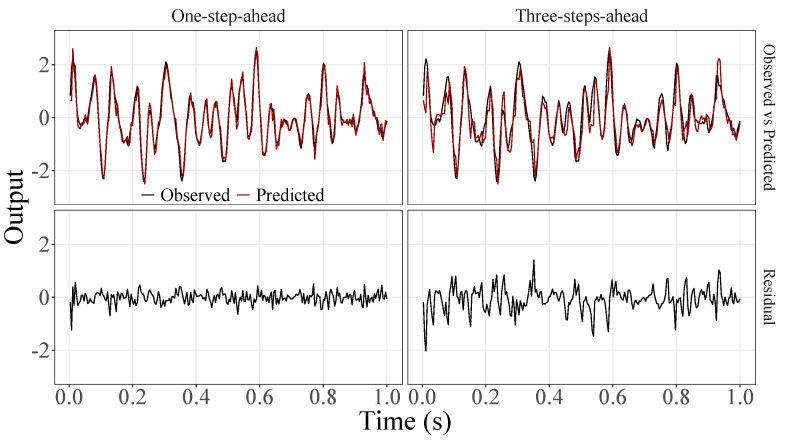
Observed EEG signals and predictions with proposed JADE-STACK model for participant 1.

**Figure 6 sensors-23-07049-f006:**
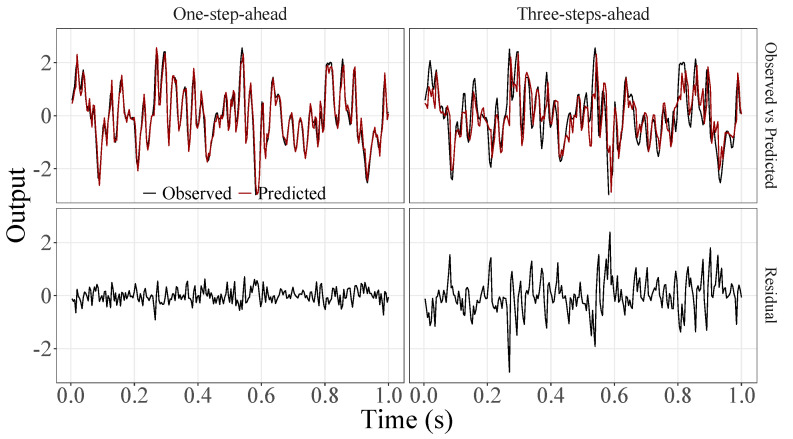
Observed EEG signals and predictions with proposed JADE-STACK model for participant 2.

**Figure 7 sensors-23-07049-f007:**
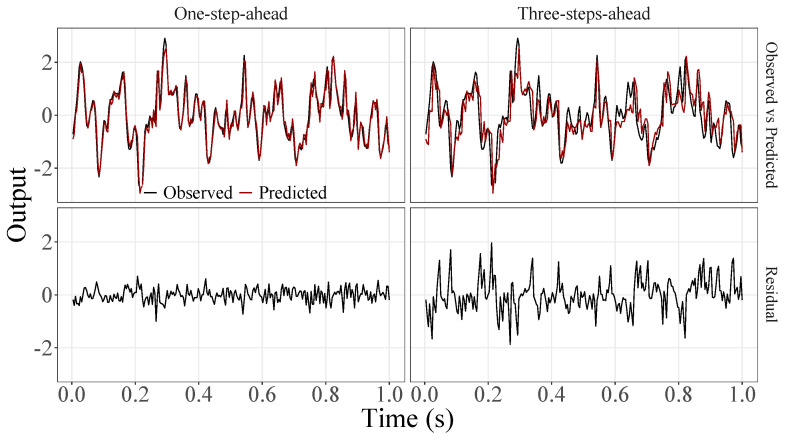
Observed EEG signals and predictions with proposed JADE-STACK model for participant 3.

**Figure 8 sensors-23-07049-f008:**
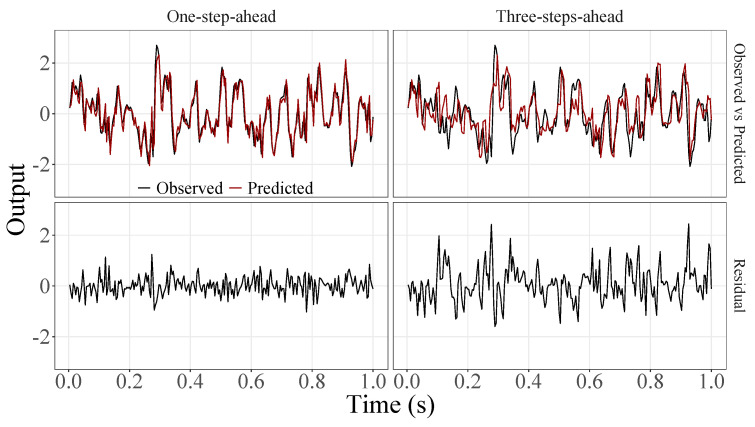
Observed EEG signals and predictions with proposed JADE-STACK model for participant 4.

**Figure 9 sensors-23-07049-f009:**
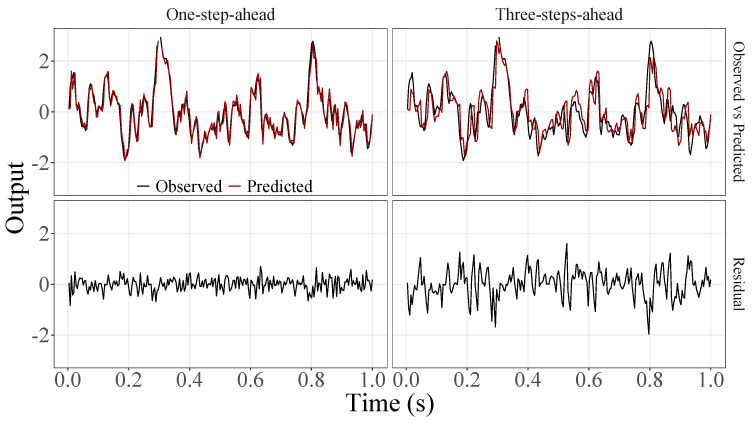
Observed EEG signals and predictions with proposed JADE-STACK model for participant 5.

**Figure 10 sensors-23-07049-f010:**
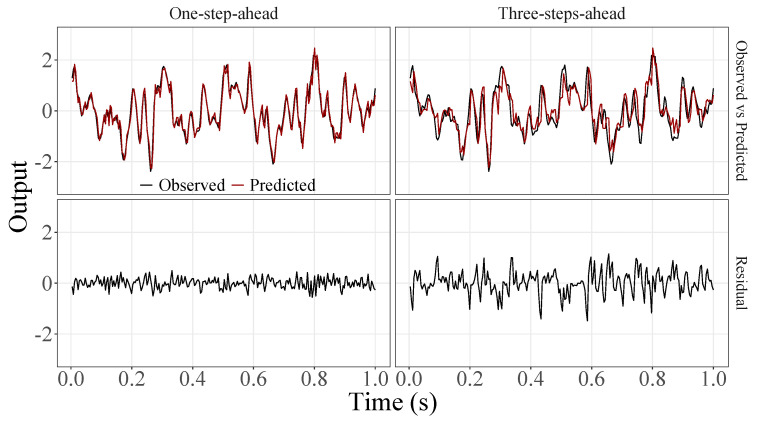
Observed EEG signals and predictions with proposed JADE-STACK model for participant 6.

**Figure 11 sensors-23-07049-f011:**
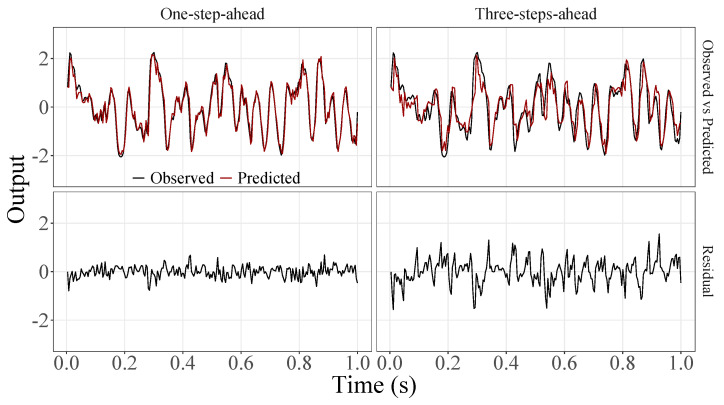
Observed EEG signals and predictions with proposed JADE-STACK model for participant 7.

**Figure 12 sensors-23-07049-f012:**
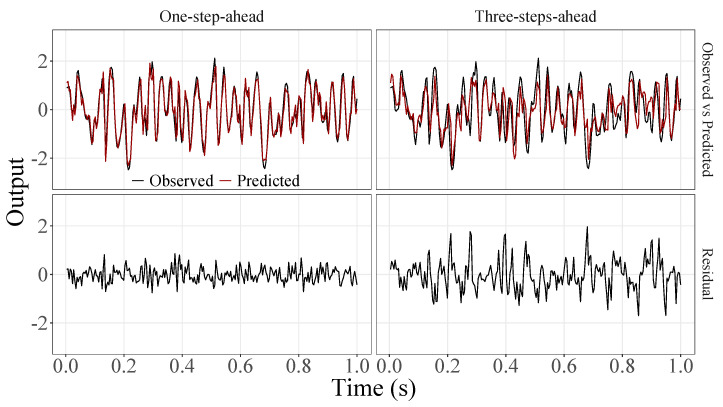
Observed EEG signals and predictions with proposed JADE-STACK model for participant 8.

**Figure 13 sensors-23-07049-f013:**
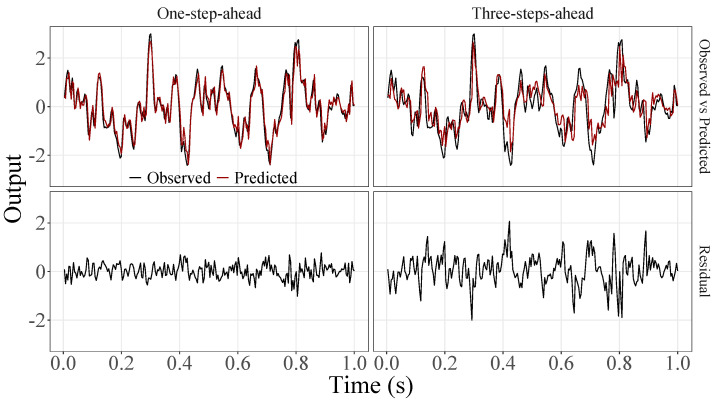
Observed EEG signals and predictions with proposed JADE-STACK model for participant 9.

**Figure 14 sensors-23-07049-f014:**
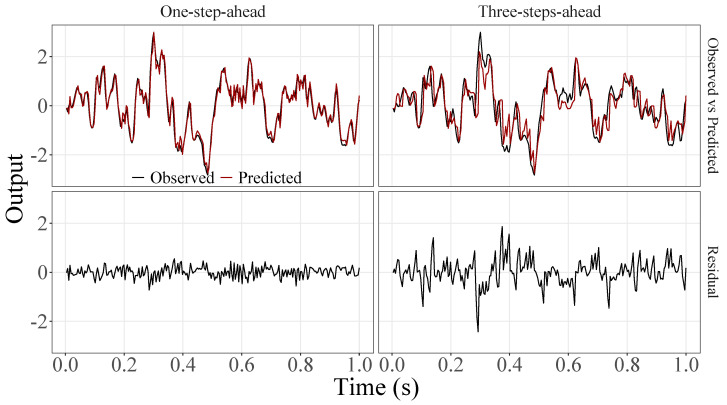
Observed EEG signals and predictions with proposed JADE-STACK model for participant 10.

**Figure 15 sensors-23-07049-f015:**
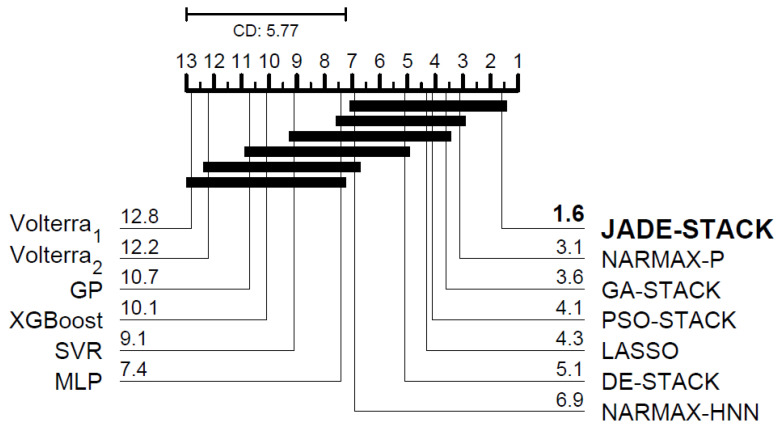
Diagram of CD for post hoc Nemenyi test for the one-step-ahead VAF.

**Figure 16 sensors-23-07049-f016:**
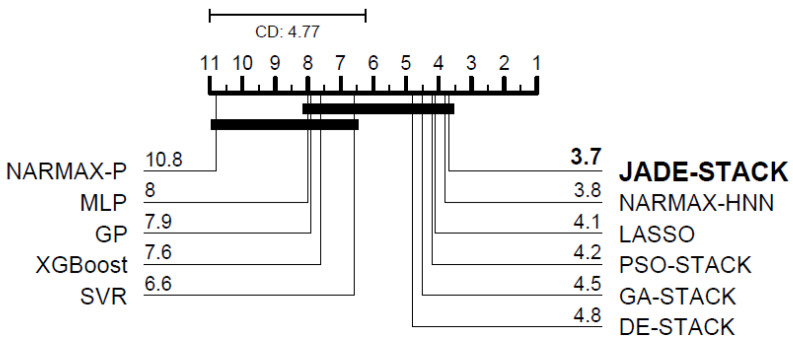
Diagram of CD for post hoc Nemenyi test for the three-step-ahead VAF.

**Table 1 sensors-23-07049-t001:** Selected hyperparameters for base learners (by grid search) and meta-learner (by optimization) for every participant from 1 to 10 in the original experiment of [37].

	Base Learners (Layer-0)									Meta-Learner (Layer-1)	
	LASSO	MLP	GP	SVR		XGBoost				Cubist	
Participant	Regularization	#Hidden Units	Sigma	Sigma	Cost	#Boosting Iterations	L2 Regularization	L1 Regularization	Learning Rate	Committees	Neighbors
1	0.9	3	0.089	0.071	1	150	0	0.1	0.3	51	0
2	0.9	5	0.075	0.066	1	150	0.1	0.1	0.3	29	1
3	0.9	5	0.074	0.07	1	150	0.1	0.1	0.3	16	1
4	0.9	5	0.073	0.068	1	150	0.1	0.0001	0.3	30	1
5	0.9	5	0.085	0.078	1	150	0.1	0.1	0.3	16	1
6	0.9	5	0.077	0.077	1	150	0.1	0.1	0.3	2	1
7	0.9	5	0.076	0.076	1	150	0.1	0.1	0.3	16	1
8	0.9	5	0.07	0.071	1	150	0.1	0.0001	0.3	0	1
9	0.9	5	0.075	0.074	1	150	0.1	0.1	0.3	4	0
10	0.9	5	0.08	0.07	1	50	0.0001	0	0.3	44	9

**Table 2 sensors-23-07049-t002:** Calculation times (in minutes) of the training process of the proposed approach.

Participant	LASSO	MLP	GP	SVR	XGBoost	JADE-STACK
1	0.0122	0.1397	0.6262	0.1592	2.3284	15.0342
2	0.0243	0.1408	0.5754	0.1576	2.2248	33.4170
3	0.0377	0.2476	0.0180	0.2561	3.6448	23.4528
4	0.0381	0.1817	0.9091	0.2936	3.0103	25.6545
5	0.0330	0.1907	0.9047	0.2678	2.9858	18.4714
6	0.0329	0.1717	0.9105	0.2506	3.0247	17.7122
7	0.0339	0.1730	0.9433	0.2372	2.9763	28.6318
8	0.0332	0.1770	0.9360	0.2489	3.0037	16.4489
9	0.0378	0.1826	0.8871	0.2470	3.0370	19.6592
10	0.0342	0.1758	0.9986	0.2641	2.9894	26.5665
Average	0.0317	0.1781	0.7709	0.2382	2.9225	22.5049
Std	0.0075	0.0282	0.2837	0.0424	0.3756	5.6899

**Table 3 sensors-23-07049-t003:** Performance measures for the proposed and compared models in the test set for one-step-ahead prediction.

	LASSO		MLP		GP		SVR		XGBoost		JADE-STACK		DE-STACK		GA-STACK		PSO-STACK		NARMAX- HNN [11]	NARMAX- P [11]	Volterra_1_ [10]	Volterra_2_ [10]
Participant	RMSE	VAF	RMSE	VAF	RMSE	VAF	RMSE	VAF	RMSE	VAF	RMSE	VAF	RMSE	VAF	RMSE	VAF	RMSE	VAF	VAF	VAF		
1	0.2225	94.75	0.2392	94.12	0.3298	88.52	0.2919	90.96	0.2791	91.77	0.2076	96.12	0.2122	95.01	0.2063	95.28	0.2051	95.34	94.37	95.52	38.37	45.00
2	0.2667	94.06	0.2792	93.58	0.4136	85.74	0.3781	88.10	0.3909	87.26	0.2616	94.48	0.2631	94.21	0.2629	94.22	0.2625	94.25	92.83	94.74	29.12	34.00
3	0.2650	93.21	0.2975	91.48	0.3466	88.40	0.3073	90.86	0.3251	89.81	0.2545	95.04	0.2616	93.47	0.2505	94.01	0.2525	93.90	90.95	92.95	32.18	40.00
4	0.3299	86.70	0.3565	84.59	0.4073	79.72	0.3724	83.04	0.4103	79.45	0.3336	91.59	0.3433	85.56	0.3309	86.61	0.3480	85.18	91.02	91.94	28.10	50.00
5	0.2529	93.12	0.2767	91.86	0.3144	89.39	0.2752	91.86	0.3183	89.11	0.2348	94.96	0.2504	93.24	0.2347	94.06	0.2340	94.09	92.58	94.04	53.74	56.00
6	0.2068	94.59	0.2223	93.91	0.2963	88.89	0.2620	91.31	0.2739	90.60	0.2031	95.99	0.2047	94.54	0.2043	94.55	0.2035	94.58	93.76	93.72	61.07	46.00
7	0.2226	95.25	0.2672	94.43	0.3000	91.38	0.2713	92.97	0.2933	91.75	0.2167	95.39	0.2215	95.13	0.2169	95.34	0.2247	95.04	93.08	95.73	54.30	60.00
8	0.2924	91.01	0.3454	88.51	0.3836	84.57	0.3514	87.02	0.4083	82.55	0.2950	93.10	0.3061	90.19	0.2939	90.95	0.3293	88.64	90.23	91.90	39.95	51.00
9	0.2831	92.48	0.3071	91.33	0.4144	83.97	0.3792	86.67	0.3953	85.37	0.2834	92.37	0.2949	91.90	0.2864	92.33	0.2841	92.44	90.36	92.24	26.35	36.00
10	0.2137	95.83	0.2485	95.09	0.3216	90.57	0.2787	92.94	0.2627	93.69	0.2186	95.99	0.2128	95.94	0.2129	95.95	0.2126	95.93	94.15	96.28	65.19	44.00
Average	0.2555	93.10	0.2840	91.89	0.3528	87.12	0.3167	89.57	0.3357	88.14	0.2509	94.50	0.2571	92.92	0.2500	93.33	0.2556	92.94	92.33	93.91	42.84	46.20
Std	0.0377	2.5237	0.0414	3.0615	0.0452	3.3915	0.0457	3.0731	0.0566	4.2675	0.0407	1.5273	0.0440	2.9279	0.0408	2.6421	0.0483	3.2360	1.4935	1.5458	13.7891	7.8842

**Table 4 sensors-23-07049-t004:** Performance measures for the proposed and compared models in the test set for three-step-ahead predictions.

	LASSO		MLP		GP		SVR		XGBoost		JADE-STACK		DE-STACK		GA-STACK		PSO-STACK		NARMAX- HNN [11]	NARMAX- P [11]
Participant	RMSE	VAF	RMSE	VAF	RMSE	VAF	RMSE	VAF	RMSE	VAF	RMSE	VAF	RMSE	VAF	RMSE	VAF	RMSE	VAF	VAF	VAF
1	0.4605	77.62	0.5460	70.97	0.5626	66.39	0.5514	67.95	0.5654	66.12	0.4655	77.59	0.3618	73.81	0.4610	77.81	0.4577	78.28	63.44	57.08
2	0.6822	61.16	0.6467	65.57	0.7530	52.82	0.7531	52.85	0.7760	50.13	0.6589	63.77	0.6584	63.82	0.6662	62.98	0.6638	63.35	56.85	39.53
3	0.6319	61.39	0.7430	47.20	0.5953	66.02	0.5908	66.43	0.5578	70.08	0.6023	64.92	0.6342	61.14	0.5949	65.78	0.5961	65.63	67.16	31.17
4	0.6173	53.62	0.6698	46.84	0.6528	48.00	0.6415	49.96	0.7169	37.54	0.6252	52.37	0.6571	47.47	0.6194	53.68	0.6479	48.71	74.89	32.26
5	0.5485	67.66	0.6118	60.96	0.5699	65.26	0.5538	67.05	0.6426	55.64	0.5327	69.51	0.5564	66.80	0.5340	69.34	0.5333	69.48	82.31	61.57
6	0.4700	72.10	0.5190	68.02	0.5411	63.12	0.5223	65.50	0.5706	59.68	0.4656	72.55	0.4633	72.85	0.4620	72.98	0.4555	73.81	75.55	49.18
7	0.4985	76.20	0.6660	67.53	0.5515	70.85	0.5402	72.10	0.5454	71.49	0.4930	76.74	0.5106	75.07	0.4843	67.53	0.5274	73.54	74.32	65.35
8	0.6207	59.55	0.7953	43.11	0.6779	52.03	0.6685	53.03	0.7168	46.70	0.6085	61.08	0.6486	55.87	0.6283	58.56	0.6510	55.45	43.40	32.57
9	0.6232	63.69	0.6775	57.40	0.7086	53.12	0.7034	54.30	0.6932	55.09	0.6290	63.47	0.6108	65.61	0.6334	62.90	0.6289	63.33	77.16	37.98
10	0.5348	73.92	0.6664	66.34	0.6008	67.07	0.5955	67.69	0.5304	74.32	0.5445	73.03	0.5353	73.84	0.5379	73.65	0.5363	73.71	78.44	64.21
Average	0.5688	66.69	0.6541	59.39	0.6214	60.47	0.6120	61.69	0.6315	58.68	0.5625	67.50	0.5637	65.63	0.5621	66.52	0.5698	66.53	69.3520	47.09
Std	0.0727	7.6288	0.0779	9.6891	0.0690	7.6575	0.0730	7.7109	0.0840	11.3155	0.0680	7.4431	0.0931	8.5171	0.0723	6.9528	0.0746	8.7289	11.2907	13.2842

## Data Availability

The dataset used in the experiments described in this paper is confidential.

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
