# Peer review of "Decoding Electroencephalography Signal Response by Stacking Ensemble Learning and Adaptive Differential Evolution"

_sensors, 2023, doi:10.3390/s23167049_

Round 1
Reviewer 1 Report
The manuscript presents an interesting approach to Decoding Electroencephalography Signal Response by Stacking Ensemble Learning and Adaptive Differential Evolution. However, I am suggesting some comments/Suggestions for authors.
1. Add some qualtattive data in abstract like how much accuracy and stability improved Lack of detailed discussion on the specific challenges faced in identifying neural data using EEG signals:
2. The abstract briefly mentions the challenges of identifying neural data, but it would be beneficial to provide more specific details about the difficulties associated with EEG signal analysis, such as noise, artifacts, and the nonlinearity of brain dynamics.
3. The manuscript needs to be better structured if possible , with clear headings and subheadings that guide the reader through the research process.
4. I have founded some grammatical errors , kindly proof read the manuscript.
5. Does your proposed algo work with fault tolerance .
6. improve the images quality if possible .
7. Limited information on the dataset and participants.
8. Insufficient discussion on the limitations of the proposed framework in conclusion section.
9. Future research directions are mentioned but not elaborated . Authors are suggested to add details for this which will help authors to extend the research.
have some grammtical errors but english is fine
Author Response
Reviewer #1
The manuscript presents an interesting approach to Decoding Electroencephalography Signal Response by Stacking Ensemble Learning and Adaptive Differential Evolution. However, I am suggesting some comments/Suggestions for authors.
- Add some qualtattive data in abstract like how much accuracy and stability improved Lack of detailed discussion on the specific challenges faced in identifying neural data using EEG signals. The abstract briefly mentions the challenges of identifying neural data, but it would be beneficial to provide more specific details about the difficulties associated with EEG signal analysis, such as noise, artifacts, and the nonlinearity of brain dynamics.
Answer: In this reviewed version, this issue was addressed.
- The manuscript needs to be better structured if possible , with clear headings and subheadings that guide the reader through the research process.
Answer: We did not see how to change the structure of the manuscript in order to improve it. The structure core is basically Introduction, Dataset description, Theoretical background of the methods, Methodology, Results and discussion, and Conclusions. We believe this structure is pretty straightforward to match the objective of the manuscript.
- I have founded some grammatical errors , kindly proof read the manuscript.
Answer: In this reviewed version, this issue was addressed.
- Does your proposed algo work with fault tolerance .
Answer: No, it doesn’t. In this study it was proposed an approach based on optimized stacking ensemble. Indeed, the JADE algorithm was used to define the meta-learner hyperparameters. However, in future studies, we will pay attention for fault tolerance combined with the proposed structure.
- Improve the images quality if possible .
Answer: The images proposed in the manuscript were generated in high-resolution quality. Also, we believe their content presents the necessary information that complement the discussions in the manuscript.
- Limited information on the dataset and participants.
Answer: The dataset comes from a benchmark proposed by Vlaar, M.P. (https://doi.org/10.1109/TNSRE.2017.2751650). Our study intended to briefly describe the dataset in order to focus on the experiment itself. In the manuscript we advise the readers that, for more details about the dataset, the given reference should be consulted.
- Insufficient discussion on the limitations of the proposed framework in conclusion section.
Answer: In this reviewed version, this issue was addressed.
- Future research directions are mentioned but not elaborated . Authors are suggested to add details for this which will help authors to extend the research.
Answer: In this reviewed version, this issue was addressed.
Reviewer 2 Report
The paper presents an approach to the nonlinear system identification problem on an example of Electroencephalography Signal Response. The research is well developed with vast literature review and detailed explanation of the proposed approach work. While the testing results demonstrate the increase of the performance there is nothing said about calculation complexity and the time of the calculations with proposed approach. It is interesting if the increased metrics will cost with lagrer computational time.
Minimal proofreading required.
Author Response
The paper presents an approach to the nonlinear system identification problem on an example of Electroencephalography Signal Response. The research is well developed with vast literature review and detailed explanation of the proposed approach work. While the testing results demonstrate the increase of the performance there is nothing said about calculation complexity and the time of the calculations with proposed approach. It is interesting if the increased metrics will cost with lagrer computational time.
Answer: Thank you for the comment. In this reviewed version, this issue was addressed.
Reviewer 3 Report
Decoding Electroencephalography Signal Response by Stacking Ensemble Learning and Adaptive Differential Evolution
EEG signals often has linear and non-linear components. The authors are proposing a combined approach to identify the non-linear components better as compared to current existing methods using machine learning approaches. The authors have proposed STACK-JADE, a multistep hybrid approach that somehow does a better job.
Issues with the abstract:
1. There are non-machine learning based approaches and machine learning approaches for signal estimation that do a quite good job. In fact higher order AR models are reliable to some extend. It appears that the authors are claiming that their approach is better without a thorough investigation.
2. It is not very clear what is the main scientific problem they are solving by a very complicated approach that may or may not be applicable to all variety of EEG.
3. The authors have mentioned of using the proposed method on decoding electroencephalography signal response to wrist joint perturbation. But why? This is not an ideal use case of EEG. The scientific or clinical background is completely missing form the study.
4. It is not clear what is the purpose of one-step and three-step ahead predictions???
5. The abstract doesn’t mention how much better the proposed method outperforms as compared to already existing methods/ approached.
6. It is not clear from the abstract that it is worth performing a computationally expensive method to do signal estimations when the actual usecase is missing.
Comments:
1. What is the importance of the nonlinear components in your use case.
2. “In this context, aspects associated with these models, such as structure selection and determination, parameter estimation, and interpretability, can be stated as challenges and some disadvantages [8]. Alternatively, machine learning is a promising alternative [9].” The machine learning models specifically unsupervised ones can’t explain the reason they we were able to estimate appropriately in the used data set limiting the scope of reusing it for another dataset. I am not sure this is a universal solution, unless the authors can explain why they feel machine learning would be a better solution.
3. In the methods section the details about the dataset are missing. While the author’s mentioned the benchmark dataset, still it is not clear whtat was used in the proposed manuscript. For example, which electrodes the authors wanted to estimate, what was the sampling rate. How the experiment was conducted. A brief explanation would be very helpful.
4. The details of the preprocessing is very essential but have been omitted from the manuscript.
5. The model description section 3.2 is absolutely unnecessary. What is missing is the rationale behind why these models/ algorithms were used along with the technical details.
6. It might be very useful to readers if the authors can explain the purpose of one step and three step ahead approach and their utility.
7. Line 375: It will be useful if you can provide the quantitation on the claim that the proposed method is better in all aspects.
8. A fair chunk of data processing consideration are missing. Did the authors use all EEG channels or just few, in that case which electrodes and what is the rationale. It appears without reading a bunch of previous papers (over citation), a new reader will not be able to understand the whole set up or the rationale behind the proposed analysis approach.
9. Line 399 – 402: The claim in explainability of the proposed is vague and may not be accurate.
10. Line 412: The dataset at collection was 2000 Hz and had a significantly high resolution. Any reason the authors used the reduced dataset. Down sampling is prone to lose information specifically when the events happen in millisecond scale. It appears we are trying to solve a problem after creating them.
Overall, while analytical approach makes sense, the utility of it, and fair comparison against other established methods is missing. The proposed improvements may not be significantly different from similar approaches, but having one more method in the analytical bucket is fair. The authors have not provided any details of the time and computational cost of predicting the data. If it is too time consuming and computationally intensive to implement, then the benefits of the approach may not be there at all.
NA
Author Response
Decoding Electroencephalography Signal Response by Stacking Ensemble Learning and Adaptive Differential Evolution
EEG signals often has linear and non-linear components. The authors are proposing a combined approach to identify the non-linear components better as compared to current existing methods using machine learning approaches. The authors have proposed STACK-JADE, a multistep hybrid approach that somehow does a better job.
Issues with the abstract:
- There are non-machine learning based approaches and machine learning approaches for signal estimation that do a quite good job. In fact higher order AR models are reliable to some extend. It appears that the authors are claiming that their approach is better without a thorough investigation.
Answer: Thank you for the comments. Our objective in this study was the investigation of a machine learning model combined with an optimization method for EEG data analysis. As detailed in Eq. 8, the forecasting structure is composed of AR terms (lagged signals value) and some developed features. Also, the proposed model can be viewed as an alternative in the case of EEG signals modeling. Indeed, we desired to show the effectiveness of the stacking ensemble for EEG signal identification. Also, some comparisons with AR models were done, as we presented in Tables 3 and 4, where the proposed model achieved competitive results with NARMAX structures.
- It is not very clear what is the main scientific problem they are solving by a very complicated approach that may or may not be applicable to all variety of EEG.
Answer: The main problem addressed in this study was the identification of EEG signals using a JADE-STACKING as detailed in the introduction of the study. No one model can be used for all EEG signal identification. On the other hand, we intend to show that using machine learning models can be an alternative to NARMAX and Volterra models that are widely employed in the case of nonlinear system identification. Also, the stacking ensemble can be one more approach to perform nonlinear system identification, as in the case of data that come from EEG.
- The authors have mentioned of using the proposed method on decoding electroencephalography signal response to wrist joint perturbation. But why? This is not an ideal use case of EEG. The scientific or clinical background is completely missing form the study.
Answer: The dataset used in this study comes from a benchmark proposed by Vlaar, M.P. (https://doi.org/10.1109/TNSRE.2017.2751650). Our study intended to briefly describe the dataset in order to focus on the experiment itself (an application of machine learning methods to signal identification). In the manuscript we advise the readers that, for more details about the dataset, the given reference should be consulted.
- It is not clear what is the purpose of one-step and three-step ahead predictions???
Answer: Since the dataset comes from a benchmark, the whole idea was to propose an experiment to compare the results with other papers that used the same dataset. Also, the Tian et al (2018) considering that dynamics of brain activity is typically in the order of few milliseconds, the three-step ahead prediction estimated 12 ms ahead EEG oscillations (based on the 256 Hz sampling rate), which is sufficient for testing the horizon. In this reviewed version, this issue was addressed.
Tian, R.; Yang, Y.; van der Helm, F.C.T.; Dewald, J.P.A. A novel approach for modeling neural responses to joint perturbations using the NARMAX method and a hierarchical neural network. Frontiers in Computational Neuroscience 2018, 12. https: //doi.org/10.3389/fncom.2018.00096.
- The abstract doesn’t mention how much better the proposed method outperforms as compared to already existing methods/ approached.
Answer: In this reviewed version, this issue was addressed.
- It is not clear from the abstract that it is worth performing a computationally expensive method to do signal estimations when the actual usecase is missing.
Answer: We choose to keep the discussion about the computational expense in the methodology section.
Comments:
- What is the importance of the nonlinear components in your use case.
Answer: The nonlinear behavior of the data becomes the challenge of this study. The problem is to propose an effective model that could handle this nonlinear data as accurately as possible. Therefore, we propose a machine learning solution to handle the nonlinearity of the data.
- “In this context, aspects associated with these models, such as structure selection and determination, parameter estimation, and interpretability, can be stated as challenges and some disadvantages [8]. Alternatively, machine learning is a promising alternative [9].” The machine learning models specifically unsupervised ones can’t explain the reason they we were able to estimate appropriately in the used data set limiting the scope of reusing it for another dataset. I am not sure this is a universal solution, unless the authors can explain why they feel machine learning would be a better solution.
Answer: The main advantage of using machine learning algorithms is their ability to deal with nonlinear and non-stationary behaviors in data in general. This ability can be helpful for the model to understand patterns and behaviors in the data. Regarding this study, the proposed approach is not a universal solution; however, the hybrid framework itself (JADE plus STACK) can be easily adjusted for other related datasets with similar characteristics. The point is that this proposed approach seems to be a suitable solution for EEG signals. Further, the use of machine learning algorithms, especially supervised ones, enables users to apply eXplainable Artificial Intelligence (XAI) approaches, such as SHAP analysis, to obtain more interpretability of the features and the model learning process.
- In the methods section the details about the dataset are missing. While the author’s mentioned the benchmark dataset, still it is not clear whtat was used in the proposed manuscript. For example, which electrodes the authors wanted to estimate, what was the sampling rate. How the experiment was conducted. A brief explanation would be very helpful.
Answer: In the last paragraph of the Description of the Benchmark section, it is stated that: “For this system, the independent component of the EEG with the highest signal-to-noise ratio is used as the output at time t (y(t), t = 0, . . . , 1s), and the handle angle when the wrist is stimulated is used as a system input at time St (u(t), t = 0, . . . , 1s).” So, the input is the handle angle and the output is the independent component of the EEG with the highest signal-to-noise ratio. The sampling rate is 2048 Hz on the experiment, with the signals being resampled at 256 Hz, as stated in the first paragraph of the “Description of the Benchmark” section, and further mentioned by the reviewer. The experiment was conducted as described in the first paragraph of the "Description of the Benchmark" section, we believe that further detailing would be out of the scope of this paper, as the focus is on the proposed method, and would be a repetition of the information available in the paper that proposes the benchmark dataset. Furthermore, the “Description of the Benchmark” section was restructured, to make the main details regarding the benchmark more clear.
- The details of the preprocessing is very essential but have been omitted from the manuscript.
Answer: The dataset used in this study is pre-processed by the authors who proposed the benchmark. However, we do not present some details regarding independent component analysis and Fourier Transform in the section dedicated to explaining the dataset, however, we direct readers to the original study.
- The model description section 3.2 is absolutely unnecessary. What is missing is the rationale behind why these models/ algorithms were used along with the technical details.
Answer: The explanation behind the use of these specific methods are presented in the introduction (in the presentation of the proposed framework) and along the manuscript. Basically, the main reason is the diversity of the algorithms, which the combination of them increases the performance of the STACK (i.e., the more heterogeneous the ensemble, the more it is robust and capable of learning the patterns and behavior of the data).
- It might be very useful to readers if the authors can explain the purpose of one step and three step ahead approach and their utility.
Answer: Since the dataset comes from a benchmark, the whole idea was to propose an experiment to compare the results with other papers that used the same dataset. Also, the Tian et al (2018) considering that dynamics of brain activity is typically in the order of few milliseconds, the three-step ahead prediction estimated 12 ms ahead EEG oscillations (based on the 256 Hz sampling rate), which is sufficient for testing the horizon. In this reviewed version, this issue was addressed. In this reviewed version, this issue was addressed.
Tian, R.; Yang, Y.; van der Helm, F.C.T.; Dewald, J.P.A. A novel approach for modeling neural responses to joint perturbations using the NARMAX method and a hierarchical neural network. Frontiers in Computational Neuroscience 2018, 12. https: //doi.org/10.3389/fncom.2018.00096.
- Line 375: It will be useful if you can provide the quantitation on the claim that the proposed method is better in all aspects.
Answer: In this reviewed version, this issue was addressed.
- A fair chunk of data processing consideration are missing. Did the authors use all EEG channels or just few, in that case which electrodes and what is the rationale. It appears without reading a bunch of previous papers (over citation), a new reader will not be able to understand the whole set up or the rationale behind the proposed analysis approach.
Answer: The EEG data was preprocessed as described "Description of the Benchmark" section, in the way that the authors of the benchmark made it available. The EEG data was resampled at 256 Hz, and the independent component of the EEG with the highest signal-to-noise ratio was used as the output. The rationale behind the proposed analysis approach is to use the proposed method to estimate the input-output relationship of the system, and then use the estimated model to predict the output of the system given the input. Furthermore, the “Description of the Benchmark” section was restructured, to make the main details regarding the benchmark more clear.
- Line 399 – 402: The claim in explainability of the proposed is vague and may not be accurate.
Answer: In this reviewed version, this issue was addressed.
- Line 412: The dataset at collection was 2000 Hz and had a significantly high resolution. Any reason the authors used the reduced dataset. Down sampling is prone to lose information specifically when the events happen in millisecond scale. It appears we are trying to solve a problem after creating them.
Answer: The dataset used in this paper is a small version of the benchmark proposed by Vlaar, M.P. (https://doi.org/10.1109/TNSRE.2017.2751650) and used by Vlaar et al. (2018) and Tian et al. (2018).. This was our choice due to the processing capacity. However, further studies can be conducted using the medium and large versions of them.
Vlaar, M.P.; Birpoutsoukis, G.; Lataire, J.; Schoukens, M.; Schouten, A.C.; Schoukens, J.; Van Der Helm, F.C.T. Modeling the nonlinear cortical response in EEG evoked by wrist joint manipulation. IEEE Transactions on Neural Systems and Rehabilitation 501
Engineering 2018, 26, 205–215. https://doi.org/10.1109/TNSRE.2017.2751650. 502
Tian, R.; Yang, Y.; van der Helm, F.C.T.; Dewald, J.P.A. A novel approach for modeling neural responses to joint perturbations using the NARMAX method and a hierarchical neural network. Frontiers in Computational Neuroscience 2018, 12. https: 504
//doi.org/10.3389/fncom.2018.00096.
Overall, while analytical approach makes sense, the utility of it, and fair comparison against other established methods is missing. The proposed improvements may not be significantly different from similar approaches, but having one more method in the analytical bucket is fair. The authors have not provided any details of the time and computational cost of predicting the data. If it is too time consuming and computationally intensive to implement, then the benefits of the approach may not be there at all.
Answer: In this reviewed version, the concerns were addressed in order to improve the paper.
Round 2
Reviewer 1 Report
Accept in present form as author have done all required changes
Author Response
Accept in present form as author have done all required changes
Answer: Thank you for your valuable feedback on the revised manuscript in the previous round. We are grateful for the opportunity to address the reviewers' comments and improve the quality of our work.
Reviewer 3 Report
The authors have addressed most of the review comments. However, they have not adequately modified the manuscript to address the review concerns appropriately.
While I appreciate the competitiveness of the proposed algorithm, several details are missing as raised in the previous review comments.
NA
Author Response
The authors have addressed most of the review comments. However, they have not adequately modified the manuscript to address the review concerns appropriately. While I appreciate the competitiveness of the proposed algorithm, several details are missing as raised in the previous review comments.
Answer: Thank you for the comments. We addressed issues that we thought were relevant to the manuscript’s focus in the previous version. The objective in this study was the investigation of a machine learning model combined with an optimization method called JADE-STACKING approach for EEG data analysis. A benchmark case related to decoding electroencephalography signal response to wrist joint perturbation was approached instead of the clinical background. The experiment focus was in the comparison of the results with other papers that used the same dataset. Additionally, in the current version of the manuscript, the abstract was enhanced and the methodology section addressed computational costs. Future research will approach high-order auto regressive (AR) models in comparison with machine learning models for different ahead prediction horizons.